# Role of Luteolin-Induced Apoptosis and Autophagy in Human Glioblastoma Cell Lines

**DOI:** 10.3390/medicina57090879

**Published:** 2021-08-26

**Authors:** Hye-Sung Lee, Bong-Soo Park, Hae-Mi Kang, Jung-Han Kim, Sang-Hun Shin, In-Ryoung Kim

**Affiliations:** 1Department of Oral and Maxillofacial Surgery, School of Dentistry, Pusan National University, Yangsan-si 50612, Korea; sofun28@naver.com (H.-S.L.); ssh8080@pusan.ac.kr (S.-H.S.); 2Department of Oral Anatomy, School of Dentistry, Pusan National University, Busandaehak-ro, 49, Mulguem-eup, Yangsan-si 50612, Korea; parkbs@pusan.ac.kr (B.-S.P.); khaemi90@naver.com (H.-M.K.); 3Dental and Life Science Institute, School of Dentistry, Pusan National University, Yangsan-si 50612, Korea; 4Medical Center, Department of Oral and Maxillofacial Surgery, Dong-A University, 26, Daesingongwon-ro, Seo-gu, Busan 49201, Korea; omfsjhkim@dau.ac.kr

**Keywords:** luteolin, globlastoma, apoptosis, autophagy

## Abstract

*Background and Objectives:* Malignant glioblastoma (GBM) is caused by abnormal proliferation of glial cells, which are found in the brain. The therapeutic effects of surgical treatment, radiation therapy, and chemo-therapy against GBM are relatively poor compared with their effects against other tumors. Luteolin is abundant in peanut shells and is also found in herbs and other plants, such as thyme, green pepper, and celery. Luteolin is known to be effective against obesity and metabolic syndrome. The anti-inflammatory, and anti-cancer activities of luteolin have been investigated. Most studies have focused on the antioxidant and anti-inflammatory effects of luteolin, which is a natural flavonoid. However, the association between the induction of apoptosis by luteolin in GBM and autophagy has not yet been investigated. This study thus aimed to confirm the occurrence of luteolin-induced apoptosis and autophagy in GBM cells and to assess their relationship. *Materials and Methods*: A172 and U-373MG glioblastoma cell lines were used for this experiment. We confirmed the apoptosis effect of Luteolin on GBM cells using methods such as 3-(4,5-dimethylthiazol-2-yl)-2,5-diphenyltetrazolium bromide (MTT) assay, immunofluorescence, Flow cytometry (FACS) western blot, and real-time quantitative PCR (qPCR). *Results:* In the luteolin-treated A172 and U-373MG cells, cell viability decreased in a concentration- and time-dependent manner. In addition, in A172 and U-373MG cells treated with luteolin at concentrations greater than 100 μM, nuclear fragmentation, which is a typical morphological change characterizing apoptosis, as well as fragmentation of caspase-3 and Poly (ADP-ribose) polymerase (PARP), which are apoptosis-related factors, were observed. Autophagy was induced after treatment with at least 50 μM luteolin. Inhibition of autophagy using 3MA allowed for a low concentration of luteolin to more effectively induce apoptosis in A172 and U-373MG cells. *Conclusions:* Results showed that luteolin induces apoptosis and autophagy and that the luteolin-induced autophagy promotes cell survival. Therefore, an appropriate combination therapy involving luteolin and an autophagy inhibitor is expected to improve the prognosis of GBM treatment.

## 1. Introduction

Glioma is the most common primary tumor in the central nervous system, along with astrocytoma, glioblastoma (GBM), and oligodendrocytoma, and it occurs primarily in glial or progenitor cells [1]. Among brain tumors, GBM is a considerably malignant tumor and has a poor prognosis despite the use of extreme therapeutic interventions, including surgery, radiation therapy, combination therapy, and adjuvant chemotherapy [2]. In chemotherapy, temozolomide, doxorubicin, paclitaxel, and 5-fluorouracil are the drugs that are administered. Since these drugs only prolong the survival period rather than bring a cure among brain tumor patients, questions regarding their effectiveness have been raised [3]. The average survival period of GBM patients undergoing aggressive treatment is as short as about one year; Due to the occurrence of new mutations, alteration of epigenetic regulators, increased UPR (Unfolded protein response) due to endoplasmic reticulum stress and mitochondrial damage, avoidance of apoptosis through direct or indirect mechanisms, and continuation of autophagy, it becomes resistant to chemotherapy rapidly, making it difficult to treat [4,5,6]. Therefore, what is needed is a new therapeutic agent that inhibits the growth of brain tumors and avoids drug resistance, reducing the likelihood of recurrence and thus improving the prognosis of GBM patients.

Oxidative stress plays an important role in the pathophysiology of various types of cancer, so antioxidants have been attracting much attention as a new therapeutic strategy for cancer, and numerous researchers have been investigating this strategy [7]. Flavonoids are natural antioxidants demonstrating potent anticancer effects, and among these many flavonoids is luteolin (3,4,5,7-tetrahydroxy flavone) [8]. Luteolin exhibits various biological effects, such as anti-inflammatory, anti-allergic, and anti-cancer [9]. Numerous studies have been conducted on the anticancer effect of luteolin; luteolin has been found to demonstrate a strong anticancer activity against a series of solid tumors [10,11,12,13]. Although several studies on the anticancer effect of luteolin in GBM have been conducted, more information is still needed on luteolin-induced apoptosis [9,14,15].

In cancer research, it is important to determine whether anticancer drugs could induce cell death and through what process is cell death induced [16]. Apoptosis is cell death processes that are completely distinct from necrosis [17]. Apoptosis, called programmed cell death, is essential for development and maintenance of homeostasis in mammalian cells; it results from the breakdown of various protein components in the nucleus and cytoplasm or is caused by the sequential activation of specific enzymes, such as caspase [18]. Autophagy is a mechanism by which damaged proteins and organelles are delivered to autolysosomes for degradation and can then be reused as energy; through the above process, autophagy plays a paradoxical dual role by regulating cell survival through acquired energy sources, but conversely promoting cell death [19,20]. Moreover, autophagy plays a pivotal role in tumorigenesis including epithelial to mesenchymal transition (EMT) processes, cancer stem cell (CSC) promotion, and multidrug resistance (MDR) [5,6,20]. Several recent studies have suggested that inhibition of autophagy promotes cell death in cancer cells. Therefore, the development of anticancer drugs that can inhibit autophagy and the combination of anticancer drugs and autophagy inhibitors could be innovative methods in cancer treatment [21,22,23,24].

Although various attempts have been made to investigate the anticancer properties of luteolin against GBM, the association of apoptosis and autophagy with luteolin has not yet been investigated. Elucidating the correlation between these processes may be an important cornerstone in overcoming brain cancer. This study was thus performed to elucidate the overall apoptotic process induced by luteolin in human GBM and to confirm the role of the concurrently induced autophagy.

## 2. Materials and Methods

### 2.1. Cell Culture and Luteolin Treatment

The human GBM cell lines A172 and U-373MG were purchased from the Korean Cell Line Bank (Seoul, Korea). A172 and U-373MG cells were grown in RPMI 1640 (Hyclone, CA, USA) medium containing L-glutamine (300 mg/L), 25 mM HEPES, NaHCO_3_, and 10% fetal bovine serum (FBS, Hyclone, CA, USA). The cells were maintained in a humidified atmosphere of 5% CO_2_ incubator at 37 °C. Luteolin (Sigma, St. Louis, MO, USA) was dissolved in dimethylsulfoxide (DMSO) to prepare a 100 mM stock solution, diluted, and then used for the experiments; the working concentration ranged from 10 μM to 200 μM, and the incubation times ranged from 24 h to 72 h.

### 2.2. MTT Assay

A172 and U-373MG cells (1 × 10^4^ cells/well) were seeded in 96-well plates and incubated for 24 h to allow the cells to adhere to the wells. Both cell lines were treated with 0, 10, 25, 50, 100, and 200 μM luteolin and cultured for 24, 48, and 72 h. At the end of the set time, 10 µL of MTT (3-[4,5-dimethylthiazol-2-yl]-2,5-diphenyltetrazolium bromide; Sigma, Louis, MO, USA) solution (5 mg/mL) was added to each well and incubated for 4 h at 37 °C. The medium (mixed with MTT solution) was removed from the wells, and 100 μL DMSO was added to each well and shaken for 10 min to dissolve the formazan crystals that formed in the cells. The absorbance of the dissolved formazan crystals was measured at 570 nm using a SpectraMax iD3 microreader (BioTek, Winooski, VT, USA). All data were derived from at least three independent experiments.

### 2.3. Hoechst 33342 and JC-1 Staining

A172 and U-373MG cells (1 × 10^5^ cells/well) were seeded in 24-well plates and incubated for 24 h to allow the cells to adhere to the wells. Both cell lines were treated with 0, 100, and 200 μM luteolin for 24 h. Each well was washed three times with phosphate buffered saline (PBS) and fixed with 4% paraformaldehyde (PFA) for 10 min. The wells were washed again three times with PBS. Each well was treated with Hoechst 33342 (Sigma, St. Louis, MO, USA; in PBS; 1 μg/mL) and allowed to react for 10 min at room temperature. The morphology of the nuclei of the two cell lines was observed, photographed, and analyzed using a Lionheart FX Automated Microscope (BioTek, Winooski, VT, USA). Moreover, JC-1 (Santa Cruz Biotechnology, Santa Cruz, CA, USA) staining was performed under the same conditions and method as in Hoechst staining. JC-1 was applied at 1 µg/mL concentration. JC-red and JC-green were observed and analyzed with Lionheart FX using a rhodamine (red fluorescence) filter and a FITC (green fluorescence) filter, respectively.

### 2.4. Acridine Orange (AO) and Monodansylcadaverine (MDC) Staining

The conditions for the AO and MDC (Sigma, St. Louis, MO, USA) staining of A172 and U-373MG cells were similar to those for Hoechst staining. AO and MDC staining could reveal acidic vesicular organelles (AVOs) and autophagosomes, respectively, which were visualized using a Lionheart FX Automated Microscope (BioTek, Winooski, VT, USA). In the AO staining, whole cells showed a green fluorescence, whereas the AVOs showed a red fluorescence. Green fluorescence was detected with the FITC filter, whereas red fluorescence was detected with the rhodamine filter of the Lionheart FX Automated Microscope. In the MCD staining, the autophagosomes emitted blue fluorescence, which was detected using the DAPI filter of the Lionheart FX.

### 2.5. Immunofluorescence Staining

A172 and U-373MG cells were seeded in a Lap-Tek chamber plate (Nalge Nunc, Naperville, IL, USA) (2 × 10^4^ cells/well) and incubated for 24 h to allow the cells to adhere to the wells. Both cell lines were treated with 50, 100, and 200 μM luteolin for 24 h. Each well was washed three times with PBS and fixed with 4% PFA for 10 min. After being fixed, the cells were washed with PBS, blocked with 1% bovine serum albumin (in PBS) for 1 h, and incubated overnight at 4 °C with a primary antibody fto cleaved caspase 3 (1:100, Cell Signaling Technology, MA, USA) and LC3B (1:100, Cell Signaling Technology, MA, USA). Subsequently, the cells were washed three times with PBS for 5 min each and then incubated with a secondary antibody (1:100, FITC-conjugated; green fluorescence, Invitrogen, CA, USA) for 2 h at room temperature. The cell nuclei were stained with DAPI (blue fluorescence), and F-actin was stained with rhodamine phalloidin (red fluorescence). Each experimental group was observed and analyzed under a Zeiss LSM 750 laser scanning confocal microscope (Carl Zeiss, Göettingen, Germany).

### 2.6. Western Blot Assay

A172 and U-373MG cells (2 × 10^6^ cells/well) treated with luteolin (0–200 μM) were collected and centrifuged at 3000 rpm for 10 min to form a pellet of cells. Each pellet was lysed with 100 μL RIPA lysis buffer for 1 h at 4 °C. The lysate samples were centrifuged under a 13,000 rpm for 30 min at 4 °C. For each lysate, the amount of protein was quantified using the Bradford method, and the total amount of protein was set to 20 μg; a loading sample was then prepared. Each sample was loaded onto an SDS-PAGE gel (10%), and the protein was transferred onto a PVDF membrane. The membrane was blocked with 5% nonfat dry milk (in PBS) for 1 h and incubated overnight at 4 °C with primary antibodies to Bax (1:1000, Santa Cruz Biotechnology, CA, USA), caspase-3 (1:1000, 1:1000, Cell Signaling Technology, Beverly, MA, USA), caspase-7 (1:1000, 1:1000, Cell Signaling Technology, MA, USA), PARP (1:1000, Cell Signaling Technology, Beverly, MA, USA), ATG5 (1:1000, Cell Signaling Technology, Beverly, MA, USA), Beclin1 (1:1000, Cell Signaling Technology, Beverly, MA, USA) and LC3B(1:1000, Cell Signaling Technology, Beverly, MA, USA). The membrane was washed five times for 10 min with PBS, and the secondary antibody (Santa Cruz Biotechnology, Santa Cruz, CA, USA) was applied at a ratio of 1:5000 and incubated at room temperature for 1 h. The membrane was washed five times for 10 min with PBS and then reacted using SuperSignal West Femto (Thermo Fisher Scientific, San Jose, CA, USA) enhanced chemiluminescence substrate; the protein expression was detected using the ImageQuant LAS 500 chemiluminescence imaging system (GE Healthcare, Chicago, IL, USA).

### 2.7. Apoptosis Detection by Annexin V/PI (Propidium Iodide) Staining

A172 and U-373MG cells (1 × 106 cells) were seeded in 100 mm culture dish and incubated for 24 h to allow the cells to adhere to the dishes. Each cell was treated with 0, 50, 100, 200 μM of luteolin and cultured for 24 h. Cells were treated with trypsin and then collected and washed twice with PBS. To detect apoptosis cells, Annexin V-FITC/PI apoptosis detection kit (Enzo Life Sciences) was used. 1× binding buffer 1 mL was applied to the collected cells and incubated with 5 μL of FITC-conjugated annexin V and 5 μL of PI (propidium iodide) at room temperature for 15 min in the dark. Apoptotic cell populations were detected using a CYTOMICS FC500 instrument (Beckman Coulter, Porterville, CA, USA).

### 2.8. Real-Time PCR

The total RNA of each sample was isolated using a RNeasy mini kit (Qiagen Inc., Valencia, CA, USA); The total RNA amounts were 1 μg in each sample. The RNA from each sample was synthesized into cDNA using a RevertAid First-Strand Synthesis System kit (Thermo Fisher Scientific, Pittsburgh, PA, USA) according to the manufacturer’s protocols and as previously described [25]. Gene expression was quantified by RT-qPCR using an SYBR Green kit (Applied Biosystems, Warrington, UK) on an ABI 7500 Real-Time PCR Detection System (Applied Biosystems, Foster City, CA, USA). The primers used in this experiment were as follows: *ATG5*, forward: GCAACTCTGGATGGGATTGC, reverse: CAACTGTCCATCTGCAGCCA; *p62,* forward: GAGATTCGCCGCTTCAGCTT, reverse: GGAAAAGGCAACCAAGTCCC; *MAP1LC3B*, forward: TTCAGGTTCACAAAACCCGC, reverse: TCTCACACAGCCCGTTTACC; and *GAPDH*, forward: TCAGACACCATGGGGAAGGT, reverse: TCCCGTTCTCAGCCATGTAG. Gene expression was calculated using the comparative Ct method and quantified using the housekeeping gene GAPDH to calculate for the Ct values of all samples.

### 2.9. Statistical Analysis

All data were expressed as mean ± SD. The figures reported herein are the average of the values obtained from at least three independent experiments. Statistical analysis were performed with one-way analysis of variance (ANOVA) and Dunnett’s comparison. Differences with probability (𝑝) values of less than 0.05 were considered statistically significant.

## 3. Results

### 3.1. Luteolin Reduces the Viability of GBM Cells

First, we performed an MTT assay to determine the appropriate luteolin concentration that induces the apoptosis of the A172 and U-373MG cells. These two cell lines were treated with 0, 10, 25, 50, 100, and 200 μM luteolin, and changes in cell viability were observed for 24 h to 72 h (Figure 1). In the 24-h treatment group, the survival rates under treatments with up to 200 μM luteolin were 49.8% for the A172 cells (IC50 value was 174.28 μM ± 7.121) and 56.7% for the U373MG cells (IC50 value was 236.09 μM ± 6.334). In the treatment with 50 µM luteolin, the survival rates of the 48- and 72-h treatment groups were 48.9% and 41.3%, respectively, for the A172 cells (IC50 values were 89.84 ± 4.698 and 55.84 μM ± 5.603) and 55.3% and 49.8%, respectively, for the U-373MG cells (IC50 values were 76.80 ± 4.069 and 46.17 μM ± 4.924). These results suggested that 200 μM is a suitable luteolin concentration for a 24-h treatment and 50 μM for a treatment for 48 h or longer.

### 3.2. Luteolin Induces Apoptosis in GBM Cells

Next, we performed Hoechst staining, JC-1 staining, Western blot assay, and immunofluorescence to confirm that the cells dying due to luteolin toxicity passed through the apoptosis pathway. Compared with the control group (0 μM), the A172 and U-373MG cells treated with 100 and 200 μM luteolin displayed condensed and cleaved nuclei (Figure 2A). When apoptosis occurs through the mitochondrial intrinsic pathway, the disruption of the mitochondrial membrane potential is triggered by the changes in the Bcl-2 and bax proteins [26].

In both the luteolin-treated GBM cell lines, the mitochondrial membrane potential dropped, and the red fluorescence of JC-1 disappeared. As shown by the Western blot assay results, the luteolin treatment groups displayed a concentration-dependent decrease in the antiapoptotic protein Bcl2, and the proapoptotic protein bax tended to increase in both GBM cell lines (Figure 2B). The protein expression levels of caspase-3, caspase-7, and PARP, which are apoptosis-related proteins, were also determined. At a luteolin concentration of 100 μM or higher, procaspase-3 and procaspase-7 decreased. PARP also clearly showed fragmentation in the luteolin treatment group of 100 μM or higher. However, the form of the cleaved caspase-3 was weakly observed (Figure 3B). Thus, we determined the expression of the cleaved caspase-3 through immunofluorescence. At a luteolin concentration of 200 μM or higher, the green fluorescence of the cleaved caspase-3 was bright and deep (Figure 3A). Next, the percentage of apoptotic cell population was determined by performing annexin V/PI double staining (Figure 3C). Each of the four areas of the dot plot is represent that the Annexin V−/PI− is the live cell, the Annexin V−/PI+ is the necrosis cell, the Annexin V+/PI− is the early stage apoptosis cell and the Annexin V+/PI+ is the late stage apoptosis [27]. We confirmed that the apoptotic cell population significantly increased after luteolin treatment as follows; In the 100 μM luteolin-treated group, the apoptotic cell populations were 40.9% for A172 (early stage, 29.4%; late stage, 11.5%) and 44.4% (early, 30.1%; late, 14.3%) for U-373MG cells. In the 200 μM luteolin-treated group, the apoptotic cell populations were 49.4% for A172 (early, 31.1%; late, 18.3%) and 55.6% (early, 31.9%; late, 23.7%) for U-373MG cells. Therefore, a treatment using at least 100 μM luteolin induces the apoptosis of GBM cells via the mitochondrial intrinsic pathway.

### 3.3. Luteolin Induces Autophagy in GBM Cells

Autophagosome formation induced by luteolin was confirmed through MDC staining, whereas AVO formation was confirmed through AO staining. A172 and U-373MG cells were treated with 50 and 100 μM luteolin; after 24 h, autophagosome formation and the number of acidified organelles in the treated cells were significantly increased relative to those in the control group (Figure 4).

To evaluate the luteolin-induced autophagy process in A172 and U-373MG cells, the expression of autophagy-related proteins was investigated by immunofluorescence, Western blot assay, and real-time PCR analysis. Both cell lines were treated with 0, 25, 50, and 100 μM luteolin, and the protein expression levels of Atg5, beclin1, p62, and light chain 3 (LC3) B, which are autophagy-related proteins, were determined. The expression levels of ATG5 and beclin 1 tended to decrease following luteolin treatment. LC3-I was converted to LC3-II, and p62 slightly increased in A172 cells but showed no change in U-373MG cells (Figure 5A). Unlike its protein expression, mRNA expression of the ATG5 gene was still observed following luteolin treatment (Figure 5B). Previously, the conversion of LC3 from I to II was confirmed by Western blot analysis, and to further clarify this, the expression of LC3 was confirmed by immunofluorescence, and the group treated with luteolin showed deep and bright fluorescence of LC3 in both cell lines (Figure 5C). The results thus showed that luteolin induces apoptosis and autophagy simultaneously.

### 3.4. Luteolin-Induced Autophagy Prevents Apoptosis in GBM Cells

Finally, we used the autophagy inhibitor 3MA to investigate whether luteolin-induced autophagy in the A172 and U-373MG cell lines directly induces apoptosis and, conversely, inhibits apoptosis. All groups were divided into control, luteolin 50 μM alone, luteolin 50 μM, 3MA 1 mM combination, and 3MA μM alone. AO staining, MTT, and western blot assay were performed in these subgroups. The red fluorescence of AVOs was weaker in the 3MA monotherapy group and in the luteolin plus 3MA combination treatment group than in the luteolin alone group (Figure 6A). In both cells, cell viability was significantly reduced in the group treated with luteolin and 3MA (Figure 6B).

When the luteolin treatment alone group was compared with the luteolin and 3MA combination treatment group, the latter showed a significantly reduced LC3 protein expression and an evident PARP cleavage (Figure 6C). Therefore, luteolin induces autophagy and apoptosis simultaneously, and this time, luteolin-induced autophagy inhibits apoptosis rather than aids in the induction of apoptosis.

## 4. Discussion

Over the past few decades, a significant progress has been made in the treatment and management of cancer patients owing to the development of anticancer drugs [28]. Nevertheless, many patients still suffer due to resistance to anticancer drugs, recurrence, and various side effects of drugs [29,30]. To overcome the various negative effects of anticancer drugs, many researchers have been searching for new anticancer drugs from plant-derived compounds with fewer side effects; the efficacy of such drugs, especially those derived from flavonoid compounds, has been proven [31,32,33].

In this study, we investigated the anticancer effect of luteolin, a flavonoid, in GBM. Luteolin inhibits the proliferation of tumor cells, protects cells from tumor stimulation, activates cell cycle arrest, and induces apoptosis, thereby exerting anticancer effects against various types of cancers, such as lung, breast, prostate, and colon cancer [8]. Moreover, luteolin exerts significant effects toward GBM; luteolin reduces the migration of human GBM cell lines by inhibiting the p-IGF-1R/PI3K/AKT/mTOR signaling pathway, and it induces apoptosis through the intracellular ER stress and mitochondrial dysfunction by increasing intracellular reactive oxygen species (ROS) levels [10,34]. However, to date, no studies have established a link between luteolin-induced apoptosis and autophagy. Therefore, in this study, we investigated the correlation between luteolin-induced apoptosis and autophagy in the human GBM cell lines A172 and U-373MG. Apoptosis is an important process in various biological systems, including maintenance of homeostasis, embryonic development, and immune response to infection [35]. Apoptosis is a mechanism by which cells self-destruct in response to DNA damage. Inducing death in cancer cells through apoptosis has provided the basis for targeted therapies that can minimize the side effects of drugs toward normal cells [36]. Induction of apoptosis involves the characteristic morphological and biochemical changes in cells. Typically, DNA fragmentation occurs, resulting in nuclear splitting or activation of Bcl-3 family proteins and caspase [37]. Our results showed that in the GBM cell lines A172 and U-373MG, luteolin showed a high cytotoxicity at concentrations above 100 μM (Figure 1), and nuclear condensation and fragmentation were observed (Figure 2A). Among the Bcl-2 family proteins present in the mitochondria, Bcl-2, an antiapoptotic factor, was decreased by luteolin treatment; moreover, bax, a proapoptotic factor, was increased (Figure 2B). The changes in these two proteins caused by luteolin treatment lowered the mitochondrial membrane potential and disrupted the membrane permeability (Figure 2B). Caspase plays a role in cleaving a set of important cellular proteins to initiate apoptosis. Cytochrome c is released from mitochondria with Apaf-1 (apoptotic protease activating factor 1) and caspase-9 to form a cellular complex, which serves as a platform that promotes the activation of apoptosis [38]. The combined proteolytic activity of caspase-9 and caspase-3 cleaves various essential protein substrates, including DFF45/ICAD, lamin B, and PARP, leading to the appearance of the morphological and biochemical characteristics of apoptosis [39]. Annexin V-FITC/PI staining is a frequently used method to monitor apoptosis using a FACS instrument [40]. When apoptosis occurs, the translocation of PS (phosphatidylserine) from the inner to the outer layer of the plasma membrane occurs, exposing it from the outer surface of the cell membrane [41]. Annexin V-FITC is used as a probe with high affinity for PS, and by double staining with PI, living cells, early apoptosis, late apoptosis, and necrotic cells can be analyzed [42]. Luteolin elicited the activation of caspase protein and the fragmentation of PARP in A172 and U-373MG cells, and it was confirmed that the rate of early and late stage of apoptosis was significantly increased, especially in over 100 µM luteolin treated cells (Figure 3). Therefore, we concluded that luteolin induces apoptosis through the mitochondrial endogenous pathway in both GBM cells.

As autophagy progresses, acidification of damaged proteins and organelles occurs, and it is detected and measured through AO staining [43]. AO can emit green and red fluorescence. When AO dye is applied where autophagy occurs, it crosses the biological membrane of a cell and accumulates in the acidic compartment, resulting in red fluorescence [44]. MDC is also used as a dye to specifically stain globular autophagic vacuoles in the perinuclear region, which are a characteristic of autophagy [45]. When an acidic environment within the cell is created, autophagy essentially occurs [46]. We confirmed that AVOs accumulate and form globular autophagosomes in luteolin-treated A172 and U-373MG cells (Figure 4). During autophagy, the autophagosome uptakes cytoplasmic components, including damaged or unnecessary proteins and organelles [47]. Microtubule-related protein 1A/1B-LC3 and ATG12–ATG5 conjugation are recruited to the autophagic membrane to complete the autophagosome formation [48,49]. LC3 exists in two forms, LC3-I and LC3-II; during autophagosome formation, a part of LC3-I converts into LC3-II, which correlates with the degree of autophagosome formation [50]. Beclin-1 modulates autophagy by forming a key complex with the pre-autophagosomal structure and class III phosphoinositide 3-kinase (PI3KC3)/Vps34 [51]. Unlike LC3, which participates in the latter stages of autophagosome formation, beclin-1 participates in the early stages of autophagy, promoting the nucleation of autophagic vesicles and recruiting proteins from the cytosol [52]. Another widely used autophagy marker, p62, also known as isolate 1 (SQSTM1), is degraded solely by autophagy and serves as a marker of autophagy flux [53]. In our results, it was observed that luteolin decreased the expression levels of ATG5 and beclin 1, that LC3 switched from type I to type II, and that a slight increase or no change occurred in p62. In contrast to its protein expression, the mRNA expression of the ATG5 gene was observed following luteolin treatment (Figure 5 and Figure 6). Therefore, we conclude that luteolin clearly induces autophagy in the GBM cells. Autophagy acts like a double-edged sword (cell survival or death), it has a positive or negative effect on multidrug resistance in cancer cells [54]. Recently, many studies have reported that autophagy inhibition enhances apoptosis induced by various anticancer agents, as seen in non-small cell lung cancer, melanoma, and Myc-induced lymphoma models [55,56,57]. Therefore, autophagy inhibition is a promising therapeutic target for cancer treatment [58]. We investigated whether the inhibition of autophagy using the autophagy inhibitor 3MA could improve the luteolin-induced apoptosis. In the GBM cells, the luteolin and 3MA co-treatment resulted in accelerated apoptosis, as seen in decreased cell viability and formation of PARP fragments (Figure 6). Overall, this study revealed that luteolin induces apoptosis and autophagy simultaneously and that luteolin-induced autophagy plays a role in cell survival.

## 5. Conclusions

Treatment of glioblastoma is primarily surgical excision of the tumor, followed by radiation therapy and chemotherapy. However, to support such treatment, the patient’s condition is very important, and there are difficulties in applying surgery and anticancer drugs due to the possibility of various sequelae. In this study, we investigated whether luteolin, one of the flavonoids, has potential for the treatment of human glioblastoma. Luteolin induced apoptosis and autophagy in GBM (A172 and U373MG) cells. Here we found a very important finding, which is that inhibition of luteolin-induced autophagy promotes apoptosis. In other words, it has been found that luteolin-induced autophagy is a signal of survival and interferes with rather than aids a signal that can lead to apoptosis. However, since the results presented in this study are in vitro experiments, it is somewhat unreasonable to view autophagy inhibition by luteolin alone as a new treatment strategy. Therefore, further studies are needed and considered on the correlation between the role of luteolin in inducing autophagy, tumor growth inhibition, luteolin treatment duration, and co-culture of malignant and normal cells. Therefore, if these findings are supported, an appropriate combination therapy including an autophagy inhibitor and luteolin is expected to improve the prognosis of GBM treatment.

## Figures and Tables

**Figure 1 medicina-57-00879-f001:**
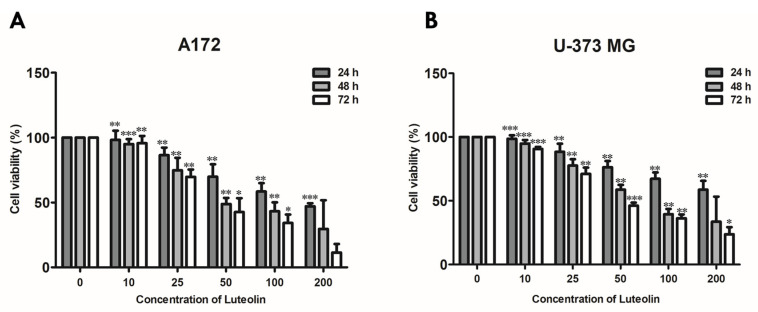
Cell viability of luteolin-treated GBM cells. A172 (**A**) and U-373MG (**B**) cells were treated with different concentrations of luteolin (0–200 μM) for 24–72 h; cell viability was measured by MTT assay. Results are expressed as mean ± SD (* *p* < 0.05, ** *p* < 0.01, and *** *p* < 0.001).

**Figure 2 medicina-57-00879-f002:**
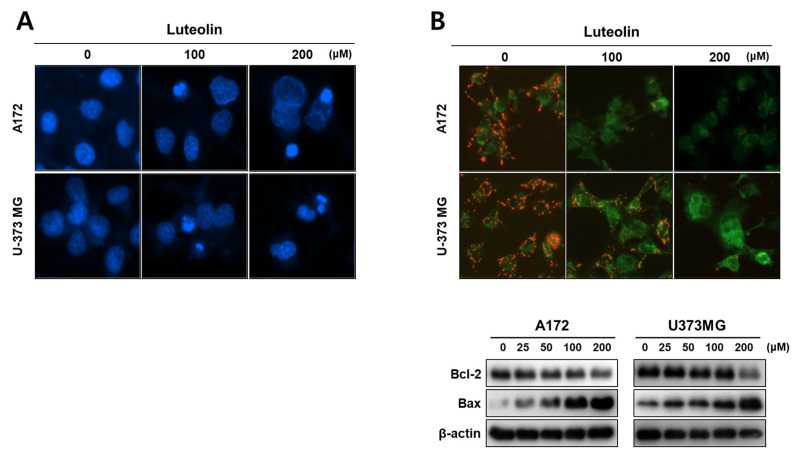
Effects of luteolin on the nucleus and mitochondria of GBM cells. (**A**) The A172 and U-373MG cells treated with luteolin (100 and 200 μM for 24 h) showed nuclear condensation and fragmentation. The nuclear morphology of the GBM cells was confirmed using Hoechst staining. (**B**) Mitochondrial membrane potential was determined by JC-1 staining (upper), and Bcl-2 and Bax expressions were determined using Western blot assay. The A172 and U-373MG cells were treated with 100 and 200 μM luteolin. In both cell lines, the JC aggregate (red fluorescence intensity) was reduced by luteolin. In the luteolin-treated cells, Bcl-2 was decreased and bax was increased in a dose-dependent manner.

**Figure 3 medicina-57-00879-f003:**
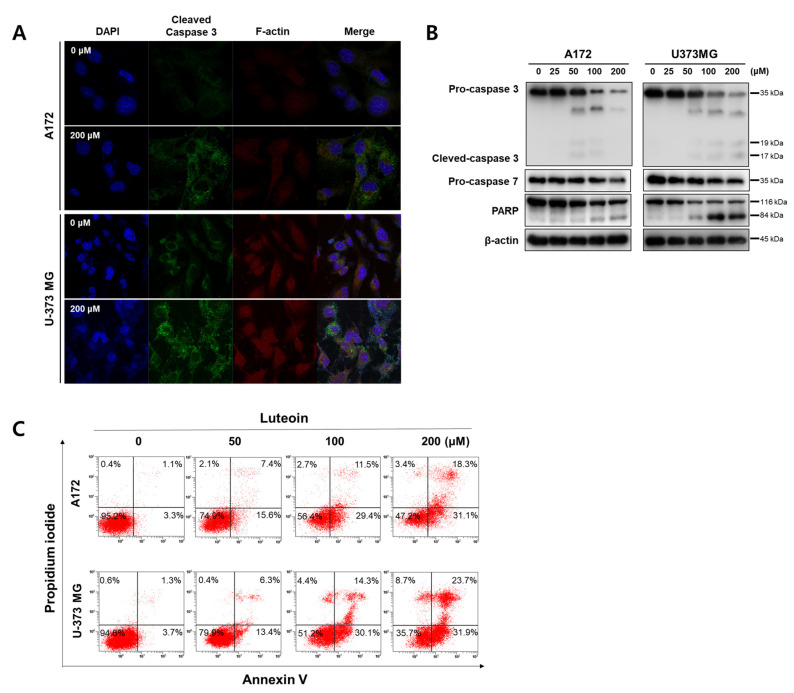
Induction of apoptosis by luteolin A172 and U-373MG cells. (**A**) Luteolin (200 μM) activates cleaved caspase-3 in both GBM cells, as confirmed by immunocytochemistry using confocal microscopy. (**B**) Luteolin (0–200 μM) treatment activated caspase-3 and caspase-7 and cleaved PARP. The protein expression levels were detected by Western blot assay. β-actin was used as internal control. **(C**) Apoptotic cell populations detect by Annexin V/PI staining in luteolin-treated GBM cells.

**Figure 4 medicina-57-00879-f004:**
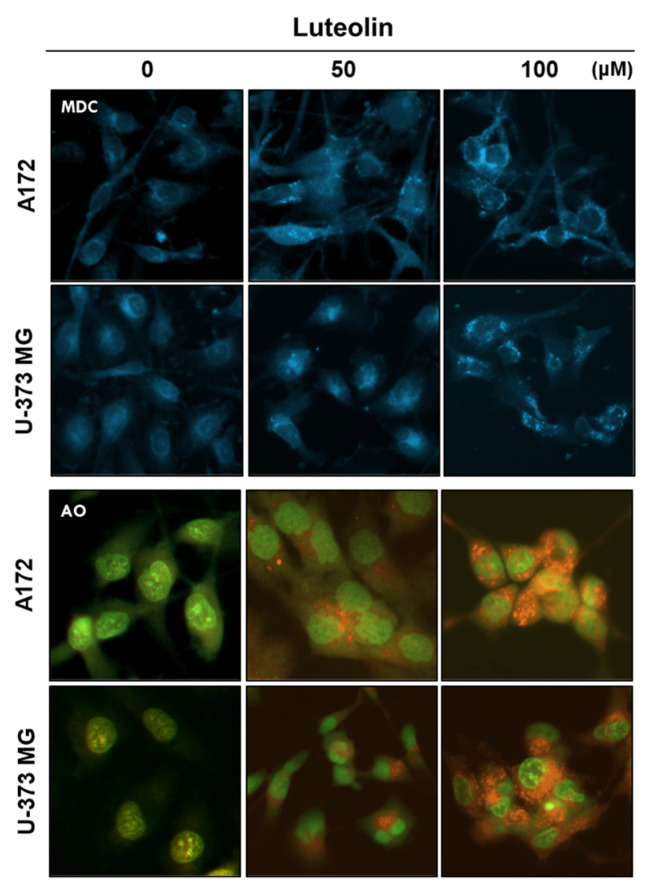
Detection of autophagic vacuoles in GBM cells using MDC and AO staining. Luteolin (50 and 100 μM) treatment induced the autophagic vacuoles in the cytosol of the A172 and U-373MG cells, as revealed by MDC staining. Luteolin (50 and 100 μM) increased the amount of AVOs (red fluorescence) in both GBM cells, as revealed by AO staining.

**Figure 5 medicina-57-00879-f005:**
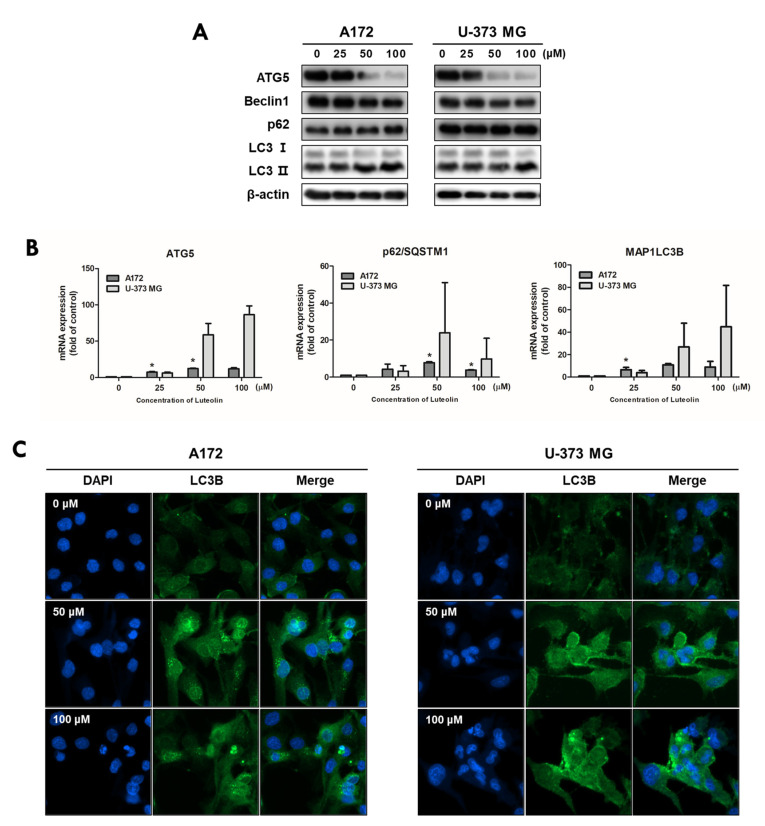
Luteolin regulates the autophagy-related factors in GBM cells. Luteolin treatment was applied at 0, 25, 50, and 100 μM; after 24 h, the expression levels of proteins and genes were confirmed by Western blot assay (**A**) and qPCR (**B**) in A172 and U-373MG cells. β-actin protein was used as internal control. (**C**) Luteolin treatments were applied at 50 and 100 μM in A172 (**A**) and U-373MG (**B**) cells, and LC3B expression was confirmed by immunofluorescence. Nuclei were visualized through DAPI staining (blue fluorescence) and LC3B staining with alexa488 (green fluorescence). The results are expressed as mean ± SD (* *p* < 0.05).

**Figure 6 medicina-57-00879-f006:**
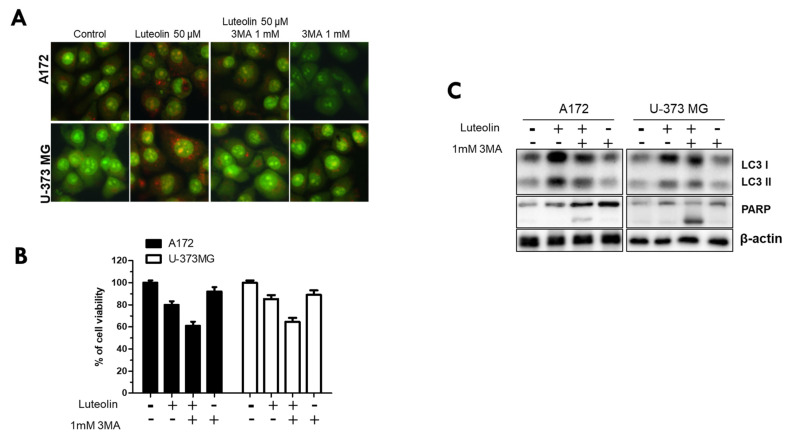
Inhibition of luteolin-induced autophagy further promotes cell death. A172 and U-373MG cells were pretreated for 1 h with 3MA (1 mM), an inhibitor of autophagy, followed by treatment with luteolin (50 µM) and then cultured for 24 h. The inhibitory effect of autophagy was confirmed by AO staining (**A**), MTT assay (**B**), and Western blot assay (**C**). β-actin was used as loading control. The results are expressed as mean ± SD.

## Data Availability

The data presented in this study are available on request from the corresponding author.

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
