# Peer review of "Role of Luteolin-Induced Apoptosis and Autophagy in Human Glioblastoma Cell Lines"

_medicina, 2021, doi:10.3390/medicina57090879_

Round 1

Reviewer 1 Report

Please, see the attached document.

Author Response

Conclusive comments:

As a main comment, I recommend to the authors to discuss more deeply the dual role of autophagy. The induction of autophagy by luteolin could perhaps be an advantage and not an effect to be counteracted. It may depend on the experimental model.

  • Thank you for your valuable comments. Based on your comments, it has been modified as follows.
  • Autophagy is a mechanism by which damaged proteins and organelles are delivered to lysosomes for degradation and can then be reused as energy; through the above process, autophagy plays a paradoxical dual role by regulating cell survival through acquired energy sources, but conversely promoting cell death. [19,20]. (line 74-76)
  • Moreover, autophagy plays a pivotal role in tumorigenesis including epithelial to mes-enchymal transition (EMT) processes, cancer stem cell (CSC) promotion, and multidrug resistance (MDR)[5,6,20]. (line 77-79)
    • Alvarez-Meythaler, et al., 10.3389/fonc.2020.586069
    • Garcia-Mayea, et al., 10.1016/j.semcancer.2019.07.022
    • Lorente et al., 10.1111/brv.12337

As a second main comment, I suggest to the authors to emphasize on the role of the exposure time and to discuss the interest a longer duration time with low doses than high doses in a short period of exposure time for further studies. This can perhaps be only possible in animal models.

  • Thank you for your valuable comments. We unfortunately do not have cell viability results at low concentrations in a longer duration time. In order to compensate for this problem, it would be nice if we could run the experiment once more according to your opinion, but it is impossible under the circumstances to execute it within 7 days, the deadline for submitting a revision thesis. We earnestly ask you to consider our situation.

As a third comment, I would like to know why the authors did not study healthy cells to compare the effects on malignant and healthy cells. The selectivity of an anticancer drug on cancer cells has to be demonstrated.

However, the effects of luteolin on the cooperation between immune and cancer cells may require other studies.

  • Thank you for your valuable comments. The starting point of this paper was whether luteolin induces apoptosis and also autophagy, and what role autophagy plays in it. As you said, I agree that comparison with normal cells is necessary. And furthermore, if the selectivity of anticancer drugs for cancer cells and animal experiments are involved, it will be reborn as an excellent thesis. However, under the current circumstances, further experiments cannot be carried out. We earnestly ask you to consider our situation.

Is luteolin an anto-oxidant as indicated in the abstract ? It does not seem obvious in the following cited reference (N°27): Wang, Q.; Wang, H.; Jia, Y.; Pan, H.; Ding, H. Luteolin induces apoptosis by ROS/ER stress and mitochondrial dysfunction in gliomablastoma. Cancer Chemother Pharmacol 2017, 79, 1031-1041, doi:10.1007/s00280-017-3299-4.

  • The reference cited in our paper mentions the oxidative stress of luteolin in Giloblastoma, but there are several studies on the antioxidant effect of luteolin.
    • Eschbendar Reddy kasala et al reported that the dietary flavone luteolin prevented B(a)P-induced lung carcinogenesis in Swiss albino rats through potent free-radical scavenging, antioxidant, anti-initiation, anti-inflammatory and anti-proliferative effects [1].
    •  Another study reported that Luteolin-induced apoptosis was accompanied by the activation of intracellular and mitochondrial reactive oxygen species scavenging through the activation of antioxidant enzymes, such as superoxide dismutase and catalase in HT-29 cells [2].
  1. Kasala, E.R.; Bodduluru, L.N.; Barua, C.C.; Gogoi, R. Antioxidant and antitumor efficacy of Luteolin, a dietary flavone on benzo (a) pyrene-induced experimental lung carcinogenesis. Biomedicine & Pharmacotherapy 2016, 82, 568-577.
  2. Kang, K.A.; Piao, M.J.; Ryu, Y.S.; Hyun, Y.J.; Park, J.E.; Shilnikova, K.; Zhen, A.X.; Kang, H.K.; Koh, Y.S.; Jeong, Y.J. Luteolin induces apoptotic cell death via antioxidant activity in human colon cancer cells. International journal of oncology 2017, 51, 1169-1178.
  • Although it is generally known that luteolin has antioxidant properties, we agree that there may be conflicts of content with the above reference, and it has been revised as follows.
    • The anti-inflammatory, and anti-cancer activities of luteolin have been investigated. Most studies have focused on the antioxidant and anti-inflammatory effects of luteolin, which is a natural flavonoid.

Due to the interaction between the production of ROS and autophagy [1] it is important to relate the effects of luteolin on autophagy with its role on the ROS production. Luteolin seems to be a prooxidant and not an antioxidant compound. The effects of luteolin may be different according to the oxidative status of the cells (normal and malignant cells).

Finally, I do not agree with the conclusion that luteolin has to be administered in combination with an autophagy inhibitor. It remains an hypothesis to be verified in other models. The final aim may not to increase apoptosis but to develop new strategies other than promoting apoptosis. Induction of autophagy could be a more promising way of therapeutic research. Increasing the duration of the treatment with luteolin could be a strategy to improve its therapeutic index. To better study the role of luteolin in inducing autophagy, and its correlation with the tumor growth inhibition, there is a need to consider other parameters, among them the duration of the treatment, the co-culture of malignant and healthy cells, the production of ROS and others that the authors could try to develop in the discussion. The role of the tumor microenvironment may also be important to understand the real anticancer effects of luteolin (via apoptosis and autophagy) and this may not be studied in cells in culture. The limits of the in vitro experiments performed by the authors must then be reported in the discussion.

Examples of references on the relationship between autophagy and oxidative stress and on the dual role of autophagy [2]:

  • Scherz-Shouval, E. Shvets, E. Fass, H. Shorer, L. Gil, Z. Elazar, Reactive oxygen species are essential for autophagy and specifically regulate the activity of Atg4, EMBO J. 26 (2007) 1749–1760. https://doi.org/10.1038/sj.emboj.7601623.
  • J. Li, Y.H. Lei, N. Yao, C.R. Wang, N. Hu, W.C. Ye, D.M. Zhang, Z.S. Chen, Autophagy and multidrug resistance in cancer, Chin. J. Cancer. 36 (2017) 52. https://doi.org/10.1186/s40880-017-0219-2.

  • As you said, it is true that it is somewhat unreasonable to argue that the inhibition of luteolin-induced autophagy as a new treatment strategy is based on the results of our tests and internal experiments. However, from the results of this study, it is an unchangeable fact that the proper inhibition of autophagy promoted apoptosis. Like your suggestion, it would be good to treat luteolin at low concentrations for a long time, and to compare with normal cells and compare with other cancer cells. However, as mentioned earlier, it is practically impossible to verify this within 7 days. Therefore, we revised discussion and conclusion section as follows (and we added the reference you suggested):
    • When an acidic environment within the cell is created, autophagy essentially occurs [43].
    • Therefore, we conclude that luteolin clearly induces autophagy in the GBM cells. Autophagy acts like a double-edged sword (cell survival or death), it has a positive or negative effect on multidrug resistance in cancer cells [51].
    • Here we found a very important finding, which is that inhibition of luteolin-induced au-tophagy promotes apoptosis. In other words, it has been found that luteolin-induced au-tophagy is a signal of survival and interferes with rather than aids a signal that can lead to apoptosis. However, since the results presented in this study are in vitro experiments, it is somewhat unreasonable to view autophagy inhibition by luteolin alone as a new treatment strategy. Therefore, further studies are needed and considered on the correlation between the role of luteolin in inducing autophagy, tumor growth inhibition, luteolin treatment duration, and co-culture of malignant and normal cells. Therefore, if these findings are supported, an appropriate combination therapy including an autophagy inhibitor and luteolin is expected to improve the prognosis of GBM treatment.

Reviewer 2 Report

Running title: Role of Luteolin-Induced Apoptosis and Autophagy in Human 2 Glioblastoma Cell Lines

The present work is part of the alternative treatments that try to use the benefits of natural compounds present in multiple plants and vegetables. The potential use of luteolin in GBM and other types of cancers reveals promising and hopeful results. Despite this, the design of the present work seems well framed in broad strokes, but on the other hand I think that it should be improved in some of the proposed experiments, as well as the writing of the work as a whole (including a thorough review of English). The main points to improve to reconsider its possible publication would be:

Abstract:

There are several incorrectly fragmented words: Lute-olin, syn-drome, con-firm, ther-apy…

Line 23, the statement “correlation” should be changed for “association”, correlation is not correct in this context.

Introduction:

Line 48, mutations are not the only cause of chemotherapy resistance, in fact, there are a lot of possible causes, among the main causes autophagy plays an important role. I invite authors to revise the following papers and also include relevant information about those.

  • Alvarez-Meythaler, et al., 10.3389/fonc.2020.586069
  • Garcia-Mayea, et al., 10.1016/j.semcancer.2019.07.022
  • Lorente et al., 10.1111/brv.12337

Line 60, authors say “…a series of solid tumors” but only studies in human glioblastoma cell lines were cited.

Lines 64-65, Autophagy is not directly a cell death mechanism, please revise the following papers to correct this statement:

  • Alvarez-Meythaler, et al., 10.3389/fonc.2020.586069
  • Lorente et al., 10.1111/brv.12337

Lines 65-66, the statement “Apoptosis 65 is programmed cell death that is essential for survival” should be clarify.

Lines 68-74, “lysosomes” should be change by “autophagolysosomes or autolysosomes”. Autophagy have also other relevant functions in this context that author should considerer to include here, for example, the role of autophagy in chemoresistance in GBM and other models (Garcia-Mayea, et al., 10.1093/carcin/bgz080, Abad et al., 10.1074/mcp.RA118.001102).

Lines 76-77, the statement “correlation” should be changed for “association”, correlation is not correct in this context.

Lines 78-80, the phrase implies that the study was done knowing the results in advance, in the introductory chapter this form of writing is not correct.

Materials and Methods:

Line 96, “medium diluted with luteolin” is not spelled correctly

Have the authors performed at least three independent experiments for all the results here showed? Authors wrote “three replicates”, but this is different.

Line 143, specify what “high rpm” was used.

Line 147-148, specify primary and secondary antibodies and their dilutions/concentrations used. SuperSignal West Femto enhanced chemiluminescence substrate (company??)

Line 156, “the total RNA of each sample was 1 μg” should be concluded.

Lines 161-162, “The primers used in this experiment are as follows” should be all in past. This problem should be improved all over the paper.

Line 172: “Statistical analysis was” plural?

Results

Can the authors explain to me what treatment the control conditions received when the comparisons with Luteolin were made? Nothing was applied to them? Was DMSO or another solvent applied?

Results 3.1.

Line 181, correct “Figrue 1”

How do the authors know with an MTT trial that this death is from apoptosis? This may be true for some concentrations and times, but it is certainly not true for all cases. Have the authors previously calculated the IC50 for luteolin on a logarithmic scale for both cell lines? Could you show such graphics?

On the other hand, in addition to the%, the SD values should be shown.

On the basis of what do the authors suggest that 200 μM at 24h and 50 μM at 48h or more are the appropriate doses of luteolin? I think it would have been interesting to show data for higher doses (eg 400 μM) where it was shown that indeed all cells died, in this way the IC50 values must be calculated to have a complete tour of the curve in the concentration range.

Results 3.2.

Figure 2. Specify the treatment time in addition to the doses.

Figure 2 and 3, and lines 209-211 does not show any Bcl2 results. The procaspase-3 and procaspase-7 images do not allow us to observe the fragmentation of these apoptosis markers. Said images must include the lowest molecular weights so that we can observe said fragmentation of the caspases (the corresponding MW of each protein being also indicated), even though the authors say that they were only observed in caspase 7.

Line 217, the primary antibody used only recognizes cleaved caspase-3 ?? What antibody is it? And the non-cleaved caspase-3 is not detected ??

I believe that to demonstrate the effect on the induction of apoptosis it would be very important to do an Annexin V assay by cytometry and / or cell cycle.

Where are these results shown? Western blot assay confirmed the nuclear expression of LC3

Discussion

Lines 303-304, Bcl-2 is not shown in figure 2B.

Lines 313-314, Caspase fragmentation should be displayed correctly, the images shown do not show it.

Line 347, Figure 7 do not exist, it should say figure 6.

Author Response

Running title: Role of Luteolin-Induced Apoptosis and Autophagy in Human Glioblastoma Cell Lines

The present work is part of the alternative treatments that try to use the benefits of natural compounds present in multiple plants and vegetables. The potential use of luteolin in GBM and other types of cancers reveals promising and hopeful results. Despite this, the design of the present work seems well framed in broad strokes, but on the other hand I think that it should be improved in some of the proposed experiments, as well as the writing of the work as a whole (including a thorough review of English). The main points to improve to reconsider its possible publication would be:

  • I appreciate your valuable comments, and I will respond sincerely.

Abstract:

There are several incorrectly fragmented words: Lute-olin, syn-drome, con-firm, ther-apy…

  • I appreciate your comments and have made the following revise accordingly.
  • Luteolin is abundant in peanut shells and is also found in herbs and other plants, such as thyme, green pepper, and celery. Luteolin is known to be effective against obesity and metabolic The antioxidant, anti-inflammatory, and anti-cancer activities of luteolin have been investigated. Most studies have focused on the antioxidant and anti-inflammatory effects of luteolin, which is a natural flavonoid. However, the association between the induction of apoptosis by luteolin in GBM and autophagy has not yet been investigated. This study thus aimed to confirm the occurrence of luteolin-induced apoptosis and autophagy in GBM cells

Line 23, the statement “correlation” should be changed for “association”, correlation is not correct in this context.

  • However, the association between the induction of apoptosis by luteolin in GBM and autophagy has not yet been investigated.

Introduction:

Line 48, mutations are not the only cause of chemotherapy resistance, in fact, there are a lot of possible causes, among the main causes autophagy plays an important role. I invite authors to revise the following papers and also include relevant information about those.

  • Alvarez-Meythaler, et al., 10.3389/fonc.2020.586069
  • Garcia-Mayea, et al., 10.1016/j.semcancer.2019.07.022
  • Lorente et al., 10.1111/brv.12337
  • Based on your comments, the sentence has been modified and a reference has been added as follows.

The average survival period of GBM patients undergoing aggressive treatment is as short as about one year; when a new mutation occurs, treatment becomes difficult because malignant GBM cells rapidly become resistant to drugs

  • The average survival period of GBM patients undergoing aggressive treatment is as short as about one year; Due to the occurrence of new mutations, alteration of epigenetic regulators, increased UPR (Unfolded protein response) due to endoplasmic reticulum stress and mitochondrial damage, avoidance of apoptosis through direct or indirect mechanisms, and continuation of autophagy, it becomes resistant to chemotherapy rapidly, making it difficult to treat [4-6].

Line 60, authors say “…a series of solid tumors” but only studies in human glioblastoma cell lines were cited.

Accordingly, your advice, the references have been cited and added as follows.

  • [11] Luteolin, a flavonoid with potential for cancer prevention and therapy, Y Lin, R Shi, X Wang, HM Shen - Current cancer drug targets, 2008 https://doi.org/10.2174/156800908786241050
  • [12] Mechanism of metastasis suppression by luteolin in breast cancer, MT Cook - Breast Cancer: Targets and Therapy, 2018 https://doi.org/10.2147/BCTT.S144202
  • [13] Anti-carcinogenic effects of the flavonoid luteolin G Seelinger, I Merfort, U Wölfle, CM Schempp - Molecules, 2008 https://doi.org/10.3390/molecules13102628

Lines 64-65, Autophagy is not directly a cell death mechanism, please revise the following papers to correct this statement:

  • Alvarez-Meythaler, et al., 10.3389/fonc.2020.586069
  • Lorente et al., 10.1111/brv.12337
  • Accordingly, your advice, the references have been cited and added as follows.
  • Apoptosis is cell death processes that are completely distinct from necrosis [17]. (Line 67-68)

Autophagy is a mechanism by which damaged proteins and organelles are delivered to lysosomes for degradation and can then be reused as energy; autophagy may also lead to cell death [19].

  • Autophagy is a mechanism by which damaged proteins and organelles are delivered to lysosomes for degradation and can then be reused as energy; through the above process, autophagy plays a paradoxical dual role by regulating cell survival through acquired energy sources, but conversely promoting cell death. [19,20]. (line 74-76)
  • Moreover, autophagy plays a pivotal role in tumorigenesis including epithelial to mes-enchymal transition (EMT) processes, cancer stem cell (CSC) promotion, and multidrug resistance (MDR)[5,6,20]. (line 77-79)

Lines 65-66, the statement “Apoptosis 65 is programmed cell death that is essential for survival” should be clarify.

  • Accordingly, your advice , I revised it as follows:

Apoptosis is programmed cell death that is essential for survival

  • Apoptosis, called programmed cell death, is essential for development and maintenance of homeostasis in mammalian cells.

Lines 68-74, “lysosomes” should be change by “autophagolysosomes or autolysosomes”. Autophagy have also other relevant functions in this context that author should considerer to include here, for example, the role of autophagy in chemoresistance in GBM and other models (Garcia-Mayea, et al., 10.1093/carcin/bgz080, Abad et al., 10.1074/mcp.RA118.001102).

  • Autophagy is a mechanism by which damaged proteins and organelles are delivered to autolysosomes for degradation and can then be reused as energy

Lines 76-77, the statement “correlation” should be changed for “association”, correlation is not correct in this context.

  • Following your advice, I revised it as follows:
  • Although various attempts have been made to investigate the anticancer properties of luteolin against GBM, the association of apoptosis and autophagy with luteolin has not yet been investigated.

Lines 78-80, the phrase implies that the study was done knowing the results in advance, in the introductory chapter this form of writing is not correct.

  • Following your advice, I revised it as follows:

This study was thus performed to elucidate the overall apoptotic process induced by luteolin in human GBM and to confirm the role of the concurrently induced autophagy.

  • Therefore, this study aims to identify apoptosis and autophagy by luteolin and to investigate the association between both process in human GBM.

Materials and Methods:

Line 96, “medium diluted with luteolin” is not spelled correctly

  • The sentence you pointed out was deemed unnecessary in context and has been deleted.
  • Both cell lines were treated with 0, 10, 25, 50, 100, and 200 μM luteolin and cultured for 24, 48, and 72 h. Each well of a 96-well plate was filled with 100 μl of the medium diluted with luteolin.

Have the authors performed at least three independent experiments for all the results here showed? Authors wrote “three replicates”, but this is different.

  • Yes, we performed independent experiments at least three times in all experiments, and we corrected mistakes in the sentence you pointed out.
    • All data were derived from at least three independent experiments.

Line 143, specify what “high rpm” was used.

  • The lysate samples were centrifuged under a 13,000 rpm for 30 min at 4°C

Line 147-148, specify primary and secondary antibodies and their dilutions/concentrations used. SuperSignal West Femto enhanced chemiluminescence substrate (company??)

  • Following your advice, I revised it as follows:
    • ~ and incubated overnight at 4°C with primary antibodies to Bax (1:1000, Santa Cruz Biotechnology, CA, USA), procaspase-3 (1:1000, Santa Cruz Biotechnology, CA, USA), procaspase-7 (1:1000, Santa Cruz Biotechnology, CA, USA), PARP (1:1000, Cell Signaling Technology, MA, USA), ATG5 (1:1000, Cell Signaling Technology, MA, USA), Beclin1 (1:1000, Cell Signaling Technology, MA, USA) and LC3B (1:1000, Cell Signaling Technology, MA, USA). The membrane was washed five times for 10 min with PBS, and the secondary antibody (Santa Cruz Biotechnology, CA, USA) was applied at a ratio of 1:5000 and incubated at room temperature for 1 h. The membrane was washed five times for 10 min with PBS and then reacted using Super-Signal West Femto (Thermo Fisher Scientific, CA, USA) enhanced chemiluminescence substrate; the protein expression was detected using the ImageQuant LAS 500 chemilu-minescence imaging system (GE Healthcare, Chicago, IL, USA).

Line 156, “the total RNA of each sample was 1 μg” should be concluded.

  • The total RNA amounts were 1 μg in each sample

Lines 161-162, “The primers used in this experiment are as follows” should be all in past. This problem should be improved all over the paper.

  • The primers used in this experiment were as follows

Line 172: “Statistical analysis was” plural?

  • The expression 'analysis' is used in the singular. Therefore, it is correct to express it as 'Statistical analysis was~'.

Results

Can the authors explain to me what treatment the control conditions received when the comparisons with Luteolin were made? Nothing was applied to them? Was DMSO or another solvent applied?

  • The control group was the non-treated group to which nothing was applied. A stock solution of luteolin was prepared at 100 mM in DMSO, and the maximum applied concentration was 200 micromolar luteolin in cells. At this time, the maximum DMSO application concentration is 0.2% (v/v), which does not affect cell proliferation and viability.

Results 3.1.

Line 181, correct “Figrue 1”

How do the authors know with an MTT trial that this death is from apoptosis? This may be true for some concentrations and times, but it is certainly not true for all cases. Have the authors previously calculated the IC50 for luteolin on a logarithmic scale for both cell lines? Could you show such graphics?

On the other hand, in addition to the%, the SD values should be shown.

On the basis of what do the authors suggest that 200 μM at 24h and 50 μM at 48h or more are the appropriate doses of luteolin? I think it would have been interesting to show data for higher doses (eg 400 μM) where it was shown that indeed all cells died, in this way the IC50 values must be calculated to have a complete tour of the curve in the concentration range.

  • Thank you for your valuable comments. We unfortunately do not have cell viability results at high concentrations. In order to compensate for this problem, it would be nice if we could run the experiment once more according to your opinion, but it is impossible under the circumstances to execute it within 7 days, the deadline for submitting a revision thesis. We earnestly ask you to consider our situation.
  • We have not previously calculated the IC50 of luteolin for both cell lines. So this time, IC50 values for luteolin were calculated respectively. In A172 cells, the IC50 value was 89.84 μM in the luteolin-treated  48 hours group,  and  84 μM at 72 hours luteolin treated group. In U373MG cells, the IC50 value was 76.8 μM  in the luteolin-treated  48 hours group,  and  46.17 μM at 72 hours luteolin treated group. Following your advice, I added the IC50 values and standard deviations of luteolin in the manuscript.
    • In the 24-hour treatment group, the survival rates under treatments with up to 200 μM luteolin were 49.8% for the A172 cells (IC50 value was 174.28 μM 121) and 56.7% for the U373MG cells (IC50 value was 236.09 μM6.334). In the treatment with 50 µM luteolin, the survival rates of the 48- and 72-hour treatment groups were 48.9% and 41.3%, respectively, for the A172 cells (IC50 values were 89.84 4.698 and 55.84 μM5.603) and 55.3% and 49.8%, respectively, for the U-373MG cells (IC50 values were 76.80 4.069 and 46.17 μM4.924). These results suggested that 200 μM is a suitable luteolin concentration for a 24-hour treatment and 50 μM for a treatment for 48 h or longer.

Results 3.2.

Figure 2. Specify the treatment time in addition to the doses.

  • The treatment time was 24 hours and the sentence were revised as follows.
    • (A) The A172 and U-373MG cells treated with luteolin (100 and 200 μM for 24 h) showed nuclear condensation and fragmentation. (Line 215-216)

Figure 2 and 3, and lines 209-211 does not show any Bcl2 results. The procaspase-3 and procaspase-7 images do not allow us to observe the fragmentation of these apoptosis markers. Said images must include the lowest molecular weights so that we can observe said fragmentation of the caspases (the corresponding MW of each protein being also indicated), even though the authors say that they were only observed in caspase 7.

  • We made the mistake of missing data from bcl2. We kindly ask for your understanding of this situation and we have inserted the data in Figure 2.
  • We tried to confirm the caspase fragment through several replicates, but we could not find it in the western blot results. Therefore, it was confirmed that the expression of cleaved caspase 3 was increased in confocal results using the cleaved caspase3 antibody.

Line 217, the primary antibody used only recognizes cleaved caspase-3 ?? What antibody is it? And the non-cleaved caspase-3 is not detected ??

  • Non cleaved caspase 3 (procaspase-3) was confirmed by western blot and it was confirmed that there was no significant change during luteolin treatment (figure 3B). The cleaved caspase 3 antibody, we used is a product of Cell Signaling Technology, and is as follows.

I believe that to demonstrate the effect on the induction of apoptosis it would be very important to do an Annexin V assay by cytometry and / or cell cycle.

  • Thank you for your valuable comments. In previous other studies, we provided data such as cell cycle and rate analysis of apoptosis (annexin V) using FACS in the report. Unfortunately, however, an old FACS we currently have was broken and we were unable to perform annexin V analysis. We are currently planning to purchase new equipment, and we expect to be able to derive data through FACS in future research. We ask for your understanding of our situation.

Where are these results shown? Western blot assay confirmed the nuclear expression of LC3

  • I made a mistake while writing the sentence. The sentence has been received as follows:

Previously, Western blot assay confirmed the nuclear expression of LC3. Therefore, the expression of LC3 was confirmed by immunofluorescence,

  • Previously, the conversion of LC3 from I to II was confirmed by Western blot analysis, and to further clarify this, the expression of LC3 was confirmed by immunofluorescence.,

Discussion

Lines 303-304, Bcl-2 is not shown in figure 2B.

  • We made the mistake of missing data from bcl2. We kindly ask for your understanding of this situation and we have inserted the data in Figure 2.

Lines 313-314, Caspase fragmentation should be displayed correctly, the images shown do not show it.

  • As mentioned earlier, a fragment of caspase3 has not been identified. However, confocal data showed increased expression of cleaved caspase 3. We earnestly ask you to consider our situation.

Line 347, Figure 7 do not exist, it should say figure 6.

  • In the GBM cells, the luteolin and 3MA co-treatment resulted in accelerated apoptosis, as seen in decreased cell viability and formation of PARP fragments (Figure 6).

Round 2

Reviewer 2 Report

Despite being an interesting study, I consider that the authors have neglected the writing of the article a lot (forgetting to even include part of some figures). However, these errors have been largely remedied in this review. Nevertheless, I consider that there are still some results that must be correctly validated before evaluating the acceptance of this paper.

Materials and Methods:

Q: Line 96, “medium diluted with luteolin” is not spelled correctly

A: The correct sentence was not corrected

Q: Line 172: “Statistical analysis was” plural?

A: The expression 'analysis' is used in the singular. Therefore, it is correct to express it as 'Statistical analysis was~'.

Analyses is the plural form of analysis, which means there are more than one. You are considering all the analyses included in the paper, aren´t you?

Results

Results 3.1.

  • Line 181, correct “Figrue 1”, should be “Figure 1”
  • Figure 2 and 3,

Since the authors failed to fully demonstrate apoptosis by WB (because they failed to see fragmentation of the caspases), I really consider that an additional experiment in which the authors demonstrate by flow cytometry (Annexin V assay ) that effectively the concentrations of 50, 100 and 200 induce apoptosis in a dose-dependent manner.

  • Line 217, the primary antibody used only recognizes cleaved caspase-3 ?? What antibody is it? And the non-cleaved caspase-3 is not detected ??

Non cleaved caspase 3 (procaspase-3) was confirmed by western blot and it was confirmed that there was no significant change during luteolin treatment (figure 3B). The cleaved caspase 3 antibody, we used is a product of Cell Signaling Technology, and is as follows.

In Figure 3B, if we compare pro-caspase 3 at 200 uM versus control, at least for the U373MG line there is a significant decrease versus control, so you cannot be sure that your antibody had only recognize cleaved caspase-3. Therefore, this once again supports an Annexin V trial. There is usually the possibility of contracting an external service or establishing collaborations for it, it is not too complicated in my opinion.

Author Response

Thank you for your valuable comments. Your question has greatly helped increase the value of this thesis. We contacted a nearby other laboratory (Oral Oncology Lab, Oral and Maxillofacial Surgery) for assistance and performed Annexin V/PI analysis using FACS. In addition, we confirmed the expression pattern of caspase 3 by luteolin by performing Western blotting again. Once again, thank you for your interest and in-depth comments on this manuscript. Please check the attachment file.
